# The Methanol Adsorption in Microporous Alumina Agglomerates

**Grygorii Dragan, Volodymyr Kutarov \*, Mykola Poletaiev, Kyrylo Kolesnykov \***  **and Mariya Khlebnikova**

Physics Research Institute, Odesa I. I. Mechnikov National University, 2, Dvoryanska St., Odesa 65082, Ukraine; dragan@onu.edu.ua (G.D.); incomb@onu.edu.ua (M.P.); mashakh91@gmail.com (M.K.)
\* Correspondence: v.kutarov@onu.edu.ua (V.K.); kiruhauho@gmail.com (K.K.)

**Abstract:** We have studied theoretical and experimental methods of methanol adsorption in micropores of aluminum oxide agglomerates obtained by gas-dispersion synthesis. It is shown that the necessity to use strict equations based on the theory of volume filling of micropores, which implies a physical and formal analogy between the volume filling of micropores and capillary condensation, for qualitative and quantitative description of adsorption equilibrium in open slit-like micropores. The applicability of the proposed equations for the description of adsorption equilibrium in such systems is demonstrated on the example of experimental methanol adsorption/desorption isotherm on the aluminum oxide, which possesses a microporous structure.

**Keywords:** adsorption slit-like pores; aluminum oxide agglomerates; porous bodies

## 1. Introduction

Porous bodies that possess micro- and mesopores are widely used in various technological processes, in ecology and medicine in connection with the adsorption of molecular gases on the developed pore surface. In the general case the adsorption on the porous bodies should be considered as a stage-by-stage process [1]. Within pore range with a characteristic size h lower than the adsorbate molecular diameter $\sigma$, the adsorption occurs on the external body surface.

When the characteristic size of the slit-like pores increase, and the ratio of the pore width to the diameter of the adsorbate molecule is in the interval $1 < h/\sigma < 2$, the adsorption process is determined both by the interaction of the adsorbate molecules and the interaction of the adsorbate molecules with the force field of a very narrow pore. In the latter case, the potentials of the walls pores overlap. For the pores with characteristic size above, $2\sigma$ adsorption occurs via condensation [2].

On the other hand, if the pore width is in the range $\sigma < h < 2\sigma$, the adsorption process should be considered in accordance with the theory of volume filling of micropores (TVFM) [3]. However, in it was shown that the classical equations of TVFM do not exactly correspond to the basic principle of this theory, and so a new equation was proposed for describing the volume filling of cylindrical pores. Using this equation for the description of adsorption equilibrium in the slit-like pores, which are characteristic of complex agglomerated structures like aluminum combustion products, it is necessary to conduct a more detailed analysis and compare the results of calculations with experimental data.

In many practical tasks, it is necessary to know the sorption of methanol on aluminum oxide. Therefore, in this paper, the sorption of methanol by agglomerates of alumina grains obtained in a flame of laminar dust has been experimentally and theoretically investigated. Recently, the gas-dispersed synthesis method has been widely used to produce nanoparticles of oxides. Gas-dispersed flames that are formed during flaring of metal powders in a gaseous oxidizing medium are used both for scientific

purposes to study the combustion mechanisms of gas suspensions of disperse metals and for practical purposes, for example, for the synthesis of metal oxides nanopowders [4].

Earlier studies have shown that the condensed phase of combustion products includes both single nanosized metal oxide grains and larger components consisting of nanoparticle agglomerates [5–7]. Agglomerates contain a sufficiently large number of pores of different shapes and sizes, so they can be used as sorbents. The latter became attractive in connection with the development of chemical and plasm chemical technologies. However, the sorption properties of such agglomerates of metal oxides, especially aluminum oxide, have not been fully enough studied. There are only some results on the sorption properties of aluminum oxide obtained by the chemical method [8]. It is also interesting to study the sorption properties of real agglomerates obtained in a specific plasma medium, as well as methods for the theoretical description of sorption processes.

## 2. Materials and Methods

It is assumed that the process of adsorption in pores of aluminum agglomerates occurs in the same way as the volume filling process (TVFM) developed by Pierce, Wiley, Smith [9] and Dubinin [10,11], which was studied in Kutarov's works with colleagues [3]. TVFM was based on the assumption that the adsorption in the pores should be treated by volume filling (similar to the capillary condensation process) rather than by layer-by-layer filling. The physical analogy between both processes implies their formal analogy, i.e., the volume filling of pores and the capillary condensation could be expected to obey similar mathematic treatment.

All the TVFM equations currently used were derived on the basis of the Polanyi theory. The adsorption isotherm equation proposed here is also based on the Polanyi's characteristic curve [12]:

$$\theta = f(\beta A) \tag{1}$$

where, according to the TVFM, $\theta$ is the relative coverage, i.e., the ratio of the current adsorption value to its maximum possible value which corresponds to the rightmost boundary of the micropores range; note that $\theta < 1$. In Equation (1) A is the Polanyi adsorption potential which by its physical meaning determines the variation of Gibbs potential during the adsorption: $A = -\Delta G$:

$$A = RT\ln(p_0/p) \tag{2}$$

where $p$ and $p_0$ are the adsorbate pressure in the bulk phase and the saturation pressure, correspondingly, at temperature T; $\beta$ is the affinity parameter as introduced by Dubinin [10,11], R is the universal gas constant.

Various adsorption isotherm equations Dubinin-Radushkevich, Dubinin-Astakhov, Dubinin-Stoeckli [10,11] can be obtained based on the characteristic curve, using the equation of the thermodynamic perturbation theory for multilayer adsorption, see [1]:

$$\theta_n = \int_A^\infty f(\varepsilon_k)d\varepsilon_k \tag{3}$$

where $\varepsilon_k$ is the local value of the pore wall potential, $f(\varepsilon_k)$ is the fraction of adsorption space which corresponds to this $\varepsilon_k$ value. The total coverage then is:

$$\theta = \sum_{n=1}^\infty \theta_n \tag{4}$$

where $\theta_n$ is the n-th layer relative coverage. Equations (3) and (4) were used mostly to describe the multilayer adsorption. However, assuming $\theta_n = 0$ for all $n > 1$, the integration in Equation (4) with the modified Gaussian distribution for $f(\varepsilon_k)$ yields the Dubinin-Radushkevich equation [11]. If the

Weibull-Gnedenko distribution is chosen for $f(\varepsilon_k)$ then the integration in Equation (4) results in the Dubinin-Astakhov adsorption isotherm [11].

It should be stressed here that the Polanyi theory, upon which all presently known TVFM equations essentially rely, is, in fact, the special case of the Gibbs surface thermodynamics [2]. Therefore, the theoretical approach on which the TVFM adsorption isotherm is based does not involve the basic postulate of TVFM about the analogy between the volume filling and capillary condensation.

For the pore width range $\sigma < h < 2\sigma$, the quasione-dimensional phase (molecular associate, cluster which does not exhibit any surface tension) is formed in the pore.

The capillary condensation process is described by the Kelvin equation [13]:

$$x = \frac{P}{P_0} = \exp\left(-\frac{\lambda V_1}{RTt\sigma}\right) \tag{5}$$

Here $\lambda$ and $V_1$ are the surface tension and molar volume of liquid adsorbate; $\sigma$—van der Waals diameter of the adsorbate molecule; t—the thickness of the condensate film on the pore walls [10]. The governing parameter in Equation (5) is the relationship between capillary and adsorption forces [13].

In particular, the expression analogous to Kelvin's equation was proposed where the governing parameter $\varphi$ is the molecular associate energy in the potential field of pore walls. For the slit-like pores the equation derived in [14] becomes:

$$x = \frac{P}{P_0} = \exp\left\{\frac{\varphi z_0}{RTh}\left[1 - \frac{1}{2h}(\chi - 1)\right]\right\} \tag{6}$$

where h is the pore width, $z_0$ and $\chi$, $0 \leq \chi \leq 1$,—determine the geometric characteristics of the pore space, $\varphi$—potential energy of interaction of molecules in a potential field of walls. In [13,14] the parameter $z_0$ is rigorously determined analytically; however, in practical applications, its calculation is too complicated. On the other hand, within the transition region thermodynamics for low molecular substances, the approximation $h/z_0 \approx h/\sigma = \theta$ provides reasonable accuracy [13,14].

Taking into account the accepted notation, Equation (6) is represented as follows.

$$x = \frac{P}{P_0} = \exp\left\{\frac{\varphi}{RT\theta}\left[1 - \frac{1}{2h}(\chi - 1)\right]\right\} \tag{7}$$

We give a brief analysis of the Equation (7). In the general case, it is not possible to take into account the influence of the parameter $\chi$ in the Equation (7). Consider two limiting cases of the geometric characteristics of the pore space:

$$x = \frac{P}{P_0} = \exp\left\{\frac{\varphi}{RT\theta}\left[1 + \frac{1}{2h}\right]\right\} \text{ for } \chi = 0 \tag{8}$$

$$x = \frac{P}{P_0} = \exp\left\{\frac{\varphi}{RT\theta}\right\} \text{ for } \chi = 1 \tag{9}$$

However, in the framework of the thermodynamics of disordered media, the exponent in Equation (7) can be represented in the following form:

$$x = \exp\left[\left(\frac{\varphi}{RT\theta}\right)^{\alpha}\right] \tag{10}$$

For further calculations, we represent the Equation (10) in the form:

$$RT(\ln x)^{1/\alpha} = \frac{\varphi}{\theta} \tag{11}$$

The potential energy of molecular associate $\varphi$ in very thin pores was theoretically studied in [15]. In particular, the ratio of the potential energy of molecular associate in a slit-like pore to the potential energy of molecules relative to their interaction with the open plane surface $\varphi_0$ was calculated and this dependence plotted vs. h/$\sigma$ is shown by the solid curve in Figure 1.

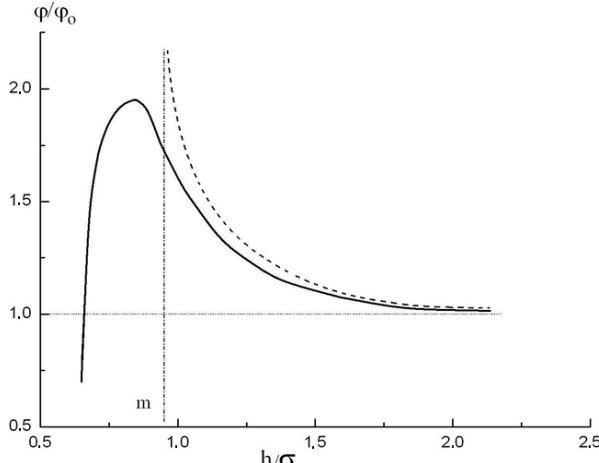

**Figure 1.** The excess of the potential energy of adsorbate molecules in a pore $\varphi$ with respect to the potential energy of molecules over the infinite plane $\varphi_0$ with relative pore size h/$\sigma$ (solid curve). Also shown are: the analytical approximation, Equation (7) (dashed curve); the schematic asymptotic of this analytical approximation at (h/$\sigma$)→m (vertical dotted line); and the asymptotic at high h/$\sigma$ (horizontal dotted line).

It is seen that the influence of the overlapping potentials created by the opposite walls remains essential even for slit-like pore width of $1.5\sigma$. The potential function shown in Figure 1 was analytically approximated in using the Kirkwood-Muller formula; however, as was mentioned in [15], it could be deduced that for practical calculations a simpler dependence shown in Figure 1 by dashed curve:

$$\overline{\varphi} = \frac{\varphi}{\varphi_0} = \frac{h/\sigma}{-m + h/\sigma} \tag{12}$$

provides a good approximation (with maximum relative error $\pm\delta = 5.5\%$) within the relative pore width range $1 < h/\sigma < 2$. Here,

$$m = 1 - \left(\frac{\varphi_0}{\varphi}\right)_{h/\sigma \to 1} \tag{13}$$

In what follows, the assumption commonly adopted in the adsorption theory is used: the relative pore width h/$\sigma$ is taken equal (within the accuracy sufficient for practical calculations) to the pore filling value $\theta$ [10], i.e., to the ratio of the current adsorption value to its value which corresponds to monolayer formation. This definition of $\theta$ is more reliable than that adopted in TVFM because the rightmost boundary of the micropores range cannot be precisely determined due to the fact that a quite large transition range exists between the micropore and mesopore regions. Then Equation (12) becomes:

$$\overline{\varphi} = \frac{\theta}{-m + \theta} \tag{14}$$

The solution of combined Equations (11) and (14) with respect to $\theta$ yields:

$$\theta = m + \frac{\varphi_0}{RT\,(\ln x)^{1/\alpha}} \tag{15}$$

Introducing the adsorption potential A = RT·ln(1/x), obtains from Equation (15):

$$\theta = m + \left(\frac{A_0}{A}\right)^{1/\alpha} \tag{16}$$

Here m $= \theta_0$—initial filling of micropores determined by the Equation (13), $A_0$ is the adsorption potential corresponding to the relative pore range where the influence of the opposite pore walls becomes negligibly small, i.e., $A_0/RT = \varphi_0/RT = -\ln(x_0)$, where $x_0$ is the relative pressure at which $\overline{\varphi} \to 1$. Therefore, Equation (16) is obtained here in the framework of the basic TVFM postulate.

Comparing Equation (12) and Figure 1 one can see that the m value is approximately equal to 1. However, in practical calculations, it is more convenient to determine the parameters in Equation (15) from the analysis of the specific isotherm considered.

## 3. Results

For an experimental study of the methanol, adsorption process on alumina agglomerates, we use results obtained by the gas-dispersed synthesis method described in [4]. Aluminum powder grade with an average diameter of 4.8 microns. After ignition, a stationary stream of combustion products was established at a temperature of 3000–3200 K, containing particles of aluminum oxide and a partially ionized gas phase. A general view of the torch is shown in Figure 2.

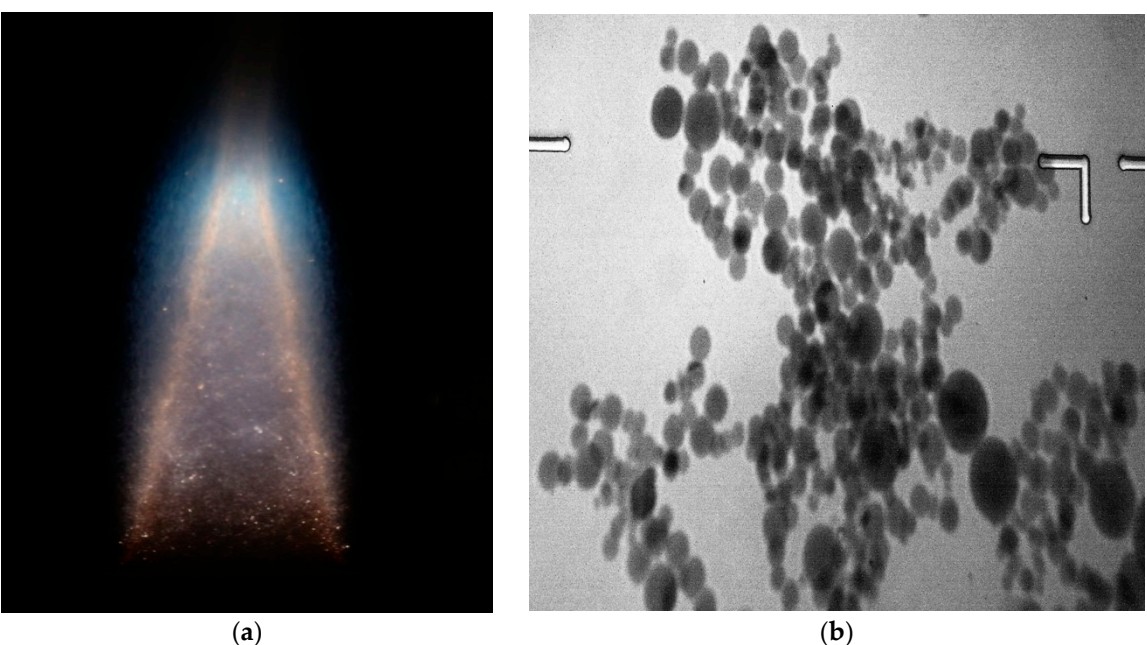

(a) (b)

**Figure 2.** Photograph of laminar diffusion dust plume of aluminum particles (**a**) and micrograph of aluminum oxide particles (**b**).

Samples of particles of aluminum oxide were photographed with an electron microscope with a magnification of 110,000. Figure 3 shows a histogram of particle size distribution.

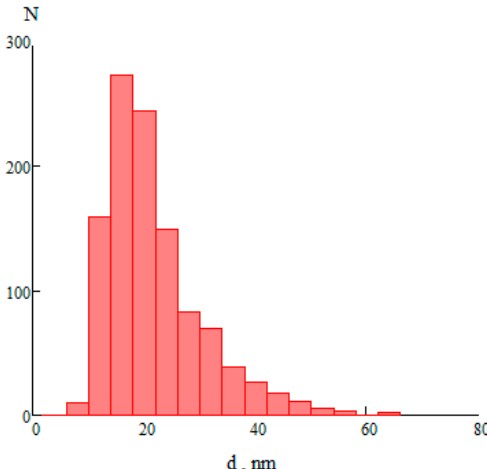

**Figure 3.** The histogram of the particle size distribution, N is the number of particles, d is the particle diameter.

The X-ray diffraction analysis showed that the particles of alumina are represented as a mixture of γ and δ-modifications.

In this connection, a number of works have been devoted to agglomeration processes, electroacoustic properties and self-ordering processes of a heterogeneous system [5–7]. It is shown, that among the combustion products the significant fraction is the agglomerates which consist of particles with dimensions about ten nanometers. Therefore, it is assumed that the agglomerates are porous bodies with an irregular pore shape, due to the association of spherical particles.

The isotherms of methanol on aluminum oxide were measured by the authors using the vacuum adsorption device with McBain-Bakr quartz helix balance at T = 294.15 K (Figure 4).

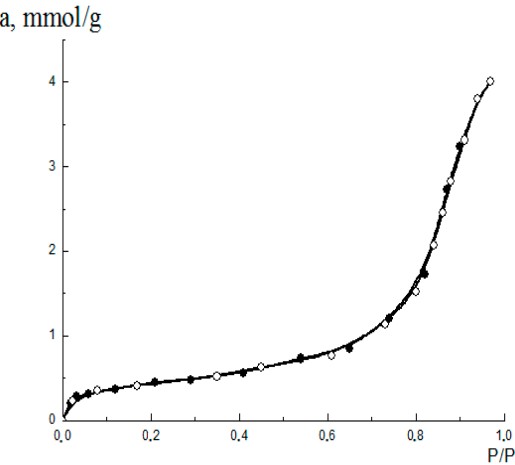

**Figure 4.** The adsorption (○)/desorption (●) isotherm for methanol vapors on the aluminum oxide material.

The initial analysis of the isotherm shown Figure 4 by its fitting to the BET Equation (2) yields the monolayer adsorption value $a_m = 0.58$ mmol/g.

For further analysis of the experimental isotherm, a t-graph was constructed (Figure 5) [11].

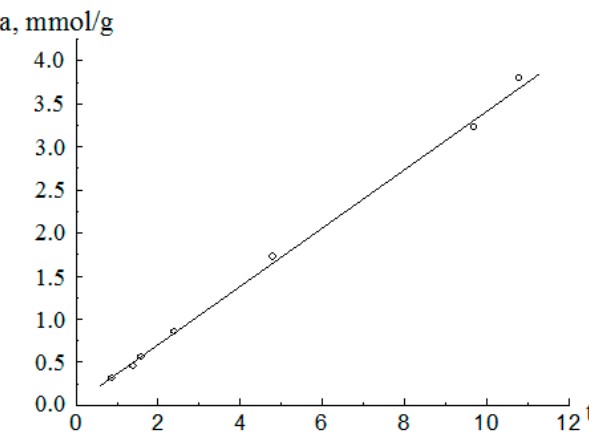

**Figure 5.** t-graf for the adsorption (○) isotherm for methanol vapors on the aluminum oxide material.

Over the entire range of variation of t, t-graph is a straight line coming from the origin. Such a pattern of the graph indicates that this material has a slit-like pore structure [16]. For slit-like pores, the microporous region is defined for the adsorption amount as $a_m \leq a < 1.6 a_m$.

To apply Equation (16) for the description of the adsorption isotherm, the isotherm was re-plotted in the coordinates

$$\ln(\theta - m) = f\left[\ln\left(A^{-1}\right)\right] \tag{17}$$

The value of m is obtained by a numerical method based on experimental dependence $\theta = f(A)^{-1}$ at $\theta \to 1$. To apply Equation (17) for the description of the adsorption isotherm, the isotherm was re-plotted in the coordinates $\theta = f(A)^{-1} \ln(\theta - m)$. The result is shown in Figure 6; it is seen that the isotherm consists of two straight intervals. For the first interval, the parameters (the m value and the slope) were obtained using the standard procedure; it was found that (to within the experimental error) m = 0.93. From Figure 6, an interesting feature of the quasi-one-dimensional adsorption regime becomes obvious: for low adsorbate density in the bulk the adsorbate in the pore exists as the molecular associate, while with the increase of the bulk adsorbate density above the value which corresponds to x = 0.85 the clustering of the associate takes place. This behavior resembles the transition from the gaseous phase to the condensed phase; however, an exact analogy with the phase transition cannot be drawn because the one-dimensional system is incapable to undergo any phase transition.

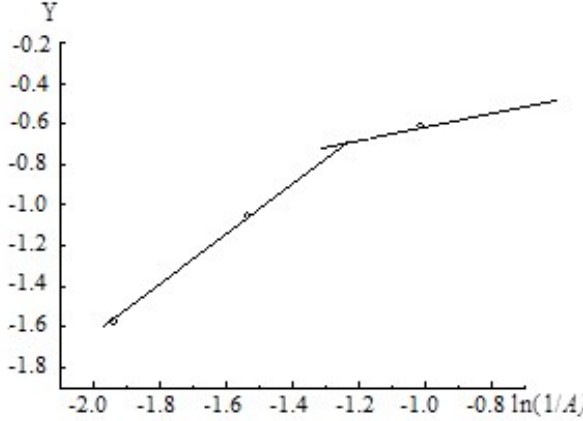

**Figure 6.** The adsorption isotherm for methanol vapors on the aluminum oxide material in the coordinates of the equation $Y = \ln(\theta - m)$.

It was noted above that the adsorption process is characterized by two stages with different process parameters. Assuming the decreasing power function as the approximation for Equation (16), one obtains for the adsorption isotherm equation:

$$\theta = 0.93 + \left(\frac{0.67}{A}\right)^{1.5} \tag{18}$$

Equation (18) describes the adsorption isotherm in the microporous region shown in Figure 4 to within the maximum relative deviation $\pm\delta = 8.7\%$.

## 4. Discussion

The Equation (15) allows us to calculate the adsorption isotherm in the region of micropores with the maximum relative deviation $\mp\delta = 8.7\%$. For comparison, we analyze the isotherm in the micropore region in the framework of the TVFM theory. In general, the equation of the adsorption isotherm in the region of micropores is written as the Dubinin-Astakhov equation [10,11]:

$$\theta_1 = \frac{a}{a_{max}} = \exp\left[-\left(\frac{A}{\varepsilon}\right)^{\alpha_1}\right] \tag{19}$$

In Equation (16), the value $a_{max}$ is defined as the maximum value of the adsorption value for the rightmost boundary in the microporous region. Unfortunately, this value is determined quite arbitrarily. In real adsorbents, the rightmost boundary in the microporous region is not clearly expressed. The value $\varepsilon$ is the characteristic energy of adsorption. The parameter $\alpha_1$ characterizes the statistical properties of the pore distribution function in terms of the adsorption energy. For a statistically homogeneous distribution (Gaussian distribution) $\alpha_1 = 2$. In this case, the Equation (16) becomes the Dubinin-Radushkevich equation. For the considered isotherm in the region of micropores, the following values of the parameters of the Equation (16) were obtained: $a_{max} = 0.58$ mmol/g; $\alpha_1 = 1.53$; $\varepsilon/RT = 3.6$. Equation (16) with the given parameters describes the adsorption isotherm in the region of micropores with the maximum relative deviation $\mp\delta = 9.1\%$.

To describe the adsorption isotherm in the micropore region, we used the classical Equation (16) and the new equation proposed by the authors [15]. Let us give some explanations and a comparative analysis of these two equations.

Equation (19) can be written in the form:

$$\theta_1 = \frac{a}{a_{max}} = \exp\left\{-\left[\left(\frac{A}{\varepsilon}\right)^2\right]^{0.765}\right\} \tag{20}$$

It can be seen from Equation (16) that the distribution of micropores in a given adsorbent should be considered as multifunctional. In this adsorbent, the entire set of micropores can be considered as a union of subsets with a Gaussian distribution.

It is interesting to note that in the Equations (15) and (16) the exponents $\alpha_1$ and $1/\alpha$ practically coincide in magnitude. In the event that the pore size distribution in the adsorbent under consideration would be statistically homogeneous, the parameter $1/\alpha$ in Equation (15) would be equal to 1.

## 5. Conclusions

Within the framework of the basic postulate involved in the TVFM, which implies the physical and formal analogy between the volume filling of micropores and the capillary condensation, the rigorous new equations are proposed for the description of adsorption equilibrium in open slit-like micropores. A feature of the use of the proposed equation for a statistically inhomogeneous pore space is shown. The determining parameters of the proposed equation in terms of physical meaning are analogous to the defining parameters of the classical equation of the TVFM theory. Of particular importance is the

fact that the control parameters of the equation proposed by the authors and the Dubinin-Astakhov equations coincide in magnitude.

So the applicability of the developed approach is illustrated by the way using the isotherm of methanol vapor adsorption on agglomerates aluminum oxide.

**Author Contributions:** G.D.—Conceptualization, Methodology, Writing—Review & Editing; V.K.—Writing—Review & Editing, Methodology, Formal Analysis; M.P.—Validation, Resources; K.K.—Investigation, Resources, Visualization; M.K.—Investigation, Resources.

**Funding:** This study was funded by the Ministry of Education and Science of Ukraine.

**Conflicts of Interest:** The authors declare no conflict of interest.

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
