# Peer review of "The Methanol Adsorption in Microporous Alumina Agglomerates"

_colloids, doi:10.3390/colloids3010022_

Round 1

Reviewer 1 Report

Minor corrections:

- in the title: microporous alumina agglomerates

- line 9: necessity

-line 27: ...above 2 sigma, adsorption occurs via condensation [2].

-line 33: Using this equation for the description of the adsorption....

-line 55: It is assumed that...

-line 58/59: treated by volume filling .....rather than by layer-by.......

-line 77: "where .... relative coverage" has to be moved to line 80 after equation 4.

-line 93: eq. 5: surface tension and thickness t are wrongly written in the equation

-line 96/97: This sentence is hard to understand

-line 103: Russian symbol has to be replaced

-line 150: The step to equation 16 is unclear corresponding to the exponent 1/alpha.

-line 155: Comparing equation (12).... not eq. (10)?

-line 160: use results....

-line 186: monolayer adsorption value a with index m is better and in Fig. 5 used

-line 196: In Fig. 5 this graphical dependence can not be seen. An additional Figure is necessary to see these results.

-line 201: micropore region .... 

Format of indices wrong:

line 94: Vl; line: 126; line 151;

Author Response

All comments of the reviewer are accepted, corrections are made in the text of the article.

Reviewer 2 Report

This manuscript has shown a study on the theoretical and experimental methods of methanol adsorption in micropores 8 of aluminum oxide agglomerates . It is shown that it is necessary to  use strict equations based on the theory of volume filling of micro pores. An experimental study was given on the adsorption and desorption curved, the both curves fit in match very well .

This might be confused as a hystersis usually exist, due to the capillary condensation. It should be explained why such hystersis was not observed.

This manuscript could be accepted after minor revision.

Author Response

Answer to Reviewer 2 comment is given in the attached file.

Reviewer 3 Report

The paper proposed a new equation for adsorption equilibrium. The reviewer has some questions and suggestions as follows.

- How can you verify that the pores in aluminum oxide agglomerates is open slit-like?

- Adsorption isotherm was measured at  T = 294.15 K only. Please explain the validity of the result in terms of temperature dependency.

- In Eq.(3), "theta" will be "theta_n"

- In Eq.(5), "lambda" will be "gamma", and one of "T" will be "t".

- "beta" in Eq.(6) is the affinity parameter used in Eq.(1)?

- Please explain what is "phi" as it is show here first.

- Please explain what is "m" in Eq.(12). 

Author Response

Answer to Reviewer 3 comment is given in the attached file. All comments of the reviewer are accepted, corrections are made in the text of the article.
